# Molecular Mechanisms, Diagnostic Aspects and Therapeutic Opportunities of Micro Ribonucleic Acids in Atrial Fibrillation

**DOI:** 10.3390/ijms21082742

**Published:** 2020-04-15

**Authors:** Allan Böhm, Marianna Vachalcova, Peter Snopek, Ljuba Bacharova, Dominika Komarova, Robert Hatala

**Affiliations:** 1National Cardiovascular Institute, 831 01 Bratislava, Slovakia; hatala@nusch.sk; 2Faculty of Medicine, Slovak Medical University, 831 01 Bratislava, Slovakia; 3Academy—Research Organization, 811 02 Bratislava, Slovakia; marianna.vachalcova@gmail.com (M.V.); pesnopek@gmail.com (P.S.); dominika.komarova@specialists.academy (D.K.); 4East-Slovak Institute of Cardiovascular Diseases, 040 11 Kosice, Slovakia; 5Cardiology Clinic Faculty Hospital, 950 01 Nitra, Slovakia; 6Saint Elisabeth University of Health and Social work, 811 02 Bratislava, Slovakia; 7Faculty of Medicine, Comenius University, 813 72 Bratislava, Slovakia; ljuba.bacharova@ilc.sk; 8International Laser Center, 841 04 Bratislava, Slovakia

**Keywords:** microRNA, atrial fibrillation, pathophysiology, biomarkers, treatment targets

## Abstract

Micro ribonucleic acids (miRNAs) are short non-coding RNA molecules responsible for regulation of gene expression. They are involved in many pathophysiological processes of a wide spectrum of diseases. Recent studies showed their involvement in atrial fibrillation. They seem to become potential screening biomarkers for atrial fibrillation and even treatment targets for this arrhythmia. The aim of this review article was to summarize the latest knowledge about miRNA and their molecular relation to the pathophysiology, diagnosis and treatment of atrial fibrillation.

## 1. Introduction

Micro ribonucleic acids (miRNAs) are short non-coding ribonucleic acids (RNAs) consisting of 21-24 nucleotides that play important role in regulation of gene expression by base-pairing with target messenger RNA (mRNA) at the posttranscriptional level [1]. The relation of miRNA to cardiovascular disease was first described in 2006 [2]. It was demonstrated that atrial fibrillation (AF) is associated with changes of miRNA levels in atrial tissue and plasma. Therefore, circulating miRNAs could serve as potential biomarkers for early identification of proarrhythmogenic substrate for AF. This might be of considerable interest especially in patients with subclinical AF [3,4]. Furthermore, targeting miRNAs could be a promising therapeutic strategy in the management of AF. The aim of this review was to summarize existing evidence and provide comprehensive insight into the role of miRNAs and their molecular relation to the pathophysiology, diagnosis and treatment of AF.

## 2. MicroRNAs

### 2.1. Types and Nomenclature

Non-coding RNAs play a pivotal role in cellular regulation. According to the length, they are divided into long non-coding RNAs and short non-coding RNAs [5]. Consisting of 21-24 nucleotides, miRNAs belong to short non-coding RNA group [6]. More than 2000 human miRNAs have currently been discovered. With the increasing number of new miRNAs, there is an increasing need to create a miRNA database. For this purpose, miRBase was created in 2002. The primary aim of this database is to assign stable and consistent names of newly discovered miRNAs. The miRBase database offers the possibility to browse all miRNAs by sequence and keywords via a web interface (http://www.mirbase.org/) [7].

The natural response to the increasing number of miRNAs was the establishment of a nomenclature system. The name usually consists of the abbreviation miR for mature miRNA, or the abbreviation mir in italics when referring to precursors or genes for a given miRNA. This abbreviation is followed by a numeral reflecting the order of their discovery. Before this naming system was established, certain miRNAs have retained non-numerical names from historical reasons, such as let-7 or lin-4. Multiple miRNAs can be evolutionary related. In order to differentiate among multiple members of the same family a letter is used after a number in the suffix (e.g., has-miR-451 and has-mir-451b) [8,9]. One miRNA can be encoded in the genome at multiple sites. To distinguish between them, a dash and a second number can be added. (e.g., miR-1-1 and miR-1-2 are same in structure, but miR-1-1 is encoded by the gene on chromosome 20 and miR-1-2 is encoded by the gene on chromosome 18). The assignment 5p and 3p represents mature miRNA sequences derived from the 5′ and 3′ arms of the mature miRNA hairpin duplex. Moreover, the species of origin is designed with a three-letter prefix (e.g., “has” for homo sapiens) [10,11].

### 2.2. Biogenesis and Function

Maturation of miRNAs is described in Figure 1. The last step of the process is production of miRNA-induced silencing complex (RISC). RISC consists of mature miRNA, trans-activation response RNA-binding protein (TRBP) and argonaut protein 2 (Ago2). The Ago2 unfolds double stranded miRNA. One strand, referred to as leader, remains in the RISC and the other is released into the cytoplasm and subsequently degraded. The generated RISC is responsible for the function of miRNAs in the post-transcriptional regulation of gene expression [12].

Only the second to eighth nucleotides of miRNA represent the seed region, which is responsible for target recognition. Target recognition is based on base complementarity: adenine binds to uracil and guanine binds to cytosine (A-U, C-G). Via binding to target mRNAs, miRNAs inhibit their translation [13]. The binding between RISC and the target mRNA has a double consequence. It causes cleavage of mRNA with subsequent degradation (if base complementarity is complete) or translation inhibition. In both cases, this will reduce the level of protein encoded by the mRNA [14]. The regulation of miRNAs is highly complex, one miRNA can target multiple mRNAs simultaneously and one mRNA can be regulated by a variety of different miRNAs [15,16]. MiRNAs regulate gene expression in order to maintain their proper function and homeostasis. They are able to affect the function and homeostasis by their excess or deficiency, which could be involved in disease development.

### 2.3. Laboratory Analysis of microRNAs

Changes in intracellular miRNA levels affect gene expression, alter intracellular signaling, and thereby significantly affect cellular metabolism, cycle and proper function. However, miRNAs can be also found outside the cell membrane, in blood and its derivatives (plasma, serum), in urine or saliva [17,18].

MiRNAs found in extracellular compartments are stable (resistant to cleavage enzymes found in plasma), their levels can be reproducibly and repeatedly determined, and most importantly for possible clinical use, their levels reflect pathophysiological events inside the organism, that is why they are now being studied as potential new biomarkers [19].

In the extracellular space, miRNAs are stored in small membrane vesicles (macrovesicles, exosomes), in apoptotic bodies, or inside high- and low-density lipoprotein particles [20]. They can be found in complexes with RNA-binding proteins, most commonly with Ago2, but also with nucleophosmin (NMP1), a multifunctional phosphoprotein with both tumor suppressor and oncogenic function, which plays a key role in ribosome formation and thus promotes cell growth and their proliferation [21,22].

Extracellular transfer of miRNA is one of the new ways of intercellular communication. In many diseases, intercellular communication is disrupted and changes in miRNA levels may reflect the presence or activity of the disease. In addition to active transport, miRNAs enter the circulation passively, for example, in tissue necrosis due to ischemia or stroke, mechanical (in surgery, trauma, arterial hypertension) or chemical (e.g., drug) damage. These miRNAs do not normally occur in blood and their presence in the circulation, therefore, reflects damage to the organs from which they originate. While most miRNAs occur in all tissues, there are also miRNAs that are overexpressed in some particular tissues. These miRNAs are referred to as tissue-specific, and these miRNAs are potential biomarkers of tissue damage [23,24,25]. Several methods measuring miRNAs extracted from tissue have been described. However, the tissue testing is very challenging for most clinical scenarios. Contrary, whole blood, plasma and serum are the standard sample types used in clinical laboratories. Therefore, their use is favorable for clinical evaluation of miRNAs. 

Quantitative real time PCR (qPCR) can be used for validation and accurate quantification of miRNAs. It is currently preferred and an easily available method that is widely used in molecular biological practice, also suitable for validating novel types of miRNAs. Other methods include microarrays and RNA-seq, which utilize high-performance capability of Next-Generation Sequencing systems [26,27]. For example, in the Slovak Republic the qPCR is available in genetic laboratories of the University hospitals and there are also many private laboratories capable of performing qPCR. The price of miRNA analysis is still approximately 6–8 times higher than the price of the traditional biomarkers, such as *N*-terminal pro-B-type natriuretic peptide (NT-proBNP), high-sensitivity cardiac troponin (cTn-hs) or Growth differentiation factor 15 (GDF-15). However, with the development of prefabricated biochemical kits the analyses are becoming cheaper. The biggest challenge today is the complicated process of the analysis itself, especially in the case of circulating miRNAs, requiring deep knowledge of the technique and experience.

## 3. Molecular Mechanisms of AF and miRNA

### 3.1. Left Atrial Remodeling 

Left atrial (LA) remodeling refers to the spectrum of pathological alterations in the electric, ionic, and molecular milieu of the LA leading to structural, mechanical and electrical changes of atrial myocardium. In the beginning, the remodeling is adaptive, while during chronic pathological stimuli it results in maladaptive effect [28,29]. Risk factors like hypertension, heart failure, diabetes mellitus, obstructive sleep apnea and obesity cause LA enlargement with a consequent decrease in LA function, which in turn promotes electrical remodeling and incident AF [30]. This process is responsible for atrial cardiomyopathy characterized as a complex of structural and functional variations, including anatomical remodeling, contractility changes, or electrophysiological alterations affecting the atrium, which can potentially lead to clinically relevant manifestations [28].

### 3.2. Association between microRNA and Structural Remodeling of LA

Structural remodeling of the atria involves changes at the level of tissue, cells and cellular organelles. The important signs of atrial remodeling are atrial stretch and atrial dilatation [31]. Atrial stretch can induce afterdepolarizations promoting focal triggered activity and may increase the atrial surface. It also shortens the refractory period and impairs conduction; thus, favoring reentrant arrhythmias [32]. On the other hand, AF persistence itself can lead to atrial dilatation (AD) and atrial wall stretch [31]. Chronic atrial dilatation can further contribute to the formation of an electroanatomic substrate by activation of numerous signaling pathways, such as angiotensin II and transforming growth factor-beta 1 (TGF-β1) leading to cellular hypertrophy, fibroblast proliferation, and tissue fibrosis [33,34].

The changes at cellular level include enlarged nuclei with dispersed chromatin, glycogen accumulation and an increase in cell size. Similarly, changes at the level of the organelles of the cell itself, namely increased number and size of mitochondria, or impaired sarcoplasmic reticulum integrity. Ultimately, this can cause degeneration to cell death of atrial myocytes, where these changes have been observed in various chronic AF models and in AF patients [35]. Structural remodeling thus appears to play an important role in the etiology of AF. 

A growing body of evidence has recently suggested the role of miRNAs, as potential molecular mediators in the structural remodeling and arrhythmogenesis of AF. Various miRNAs are important in the pathogenesis of fibrosis [36], e.g., miR-133 and miR-590 down-regulation caused by nicotine alleviated the repression of TGF-β1 and TGF-β receptor expression. Ex vivo experiments verified that transfection of miR-133 and miR-590 into cultured atrial fibroblasts decreased TGF-β1 and TGF-β receptor expression, as well as collagen content [37]. It has previously been demonstrated that miR-133a acts to regulate connective tissue growth factor (CTGF) expression as a repressor in the regulation of cardiac fibrosis. MiR-133 or miR-30c decrease CTGF expression levels to regulate structural alterations in the extracellular matrix (ECM) of the myocardium [38]. Other miRNAs that influenced myocardial fibrosis are the miR-29 family, miR-30 and miR-208. The miR-29 family is involved in the regulation of multiple target genes that participate in encoding fibrotic process such as collagens, fibrillins and elastin. It is documented that down-regulation of miR-29 induces the expression of these mRNAs, resulting in an enhanced fibrotic response [39].

MiRNA-21 and its downstream target protein known as Spry1 are involved in a process of structural remodeling via development of atrial fibrosis. This process correlates with collagen, connective tissue growth factor, lysyl oxidase and Rac-1-GTPase contents in left atrial tissue [40]. Moreover, miR-21 regulates the signal transducer and activator of transcription 3 (STAT3) pathway, through which it induces inflammation-associated atrial fibrosis [41]. STAT3 is an important regulator of cell proliferation via downstream signaling molecule of Cell adhesion molecule 1 (CADM1). CADM1 is a well-known tumor suppressor for a variety of cancers of epithelial origin. CADM1 acts through binding the receptor tyrosine kinase HER3, reducing cell proliferation [42]. In addition, miRNA-21 is one of the regulatory molecules in TGF-βinduced endothelial-to-mesenchymal transition via a PTEN/Akt-dependent pathway [43]. Therefore, it is a logical assumption that circulating levels of miRNA-21 reflect fibrotic changes that play a role in arrhythmogenic substrate for incident AF [40]. The miR-21 inhibitors can suppress cardiac fibroblasts proliferation. It can be concluded that miR-21 and CADM1 play a key role in cardiac fibrosis, indicating that miR21, CADM1 and STAT3 may serve as therapeutic targets of fibroblast activation and fibrosis [42]. Furthermore, there is a significant correlation between miR-21 serum concentration and extent of low voltage areas detected in the left atrium. A significant correlation between the relationship of miR-21 serum concentrations and 1-year outcome after catheter ablation in patients with persistent AF has been described. It could be assumed that circulating miR-21 can be useful for stratifying patients and planning the procedure (e.g., performing a pulmonary vein isolation only in patients with very low miR-21 serum concentrations or planning extensive substrate modification in patients with high miR-21 levels) [41].

The miRNA associated with structural remodeling is miR-132. It regulates the connective tissue growth factor (CTGF) gene involved in fibrosis [38]. CTGF was shown to be a promoter of extra cellular matrix (ECM) synthesis [44]. The induction of ECM synthesis occurs in parallel with the generation of fibrosis [45]. It was also documented that the expression of miR-132 was decreased, whereas the expression of CTGF increases significantly in dogs or patients with AF [38].

### 3.3. Association between microRNAs and Electrical Remodeling of LA

Atrial electrical remodeling (AER) refers the shortening of atrial action potential duration (APD), refractory period and the decrease of atrial conduction velocity [46]. The ion channels that influence action potential generation, duration and propagation are the L-type Ca^2+^ channel (Cav1.2), the voltage-gated potassium channel Kv4.2 or the G-protein-activated inwardly rectifying potassium channel (GIRK1/4) [47]. AF may be triggered due to premature pulmonary vein discharges, rapid electrical stimulation of atria, or a simple wave break. Oxidative stress as a result of sustained high-frequency excitation plays an important role in this process [48]. It causes formation of reactive oxygen species (ROS), which are products of oxidative enzymes. For example, the nicotinamide adenine dinucleotide phosphate oxidases (NOX) 2/4, resulting in a rapid (i.e., over hours or days) reduction of Ca^2+^ current L (ICa, L) and the Rectifier K^+^ inward current (IK1) increase, resulting in shortening of APD and the refractory period, supporting rotor formation and stabilization [49]. Reduction of the density of Cav1.2 is the hallmark of the electrical remodeling [50].

It is now accepted that some miRNAs are associated with AER. Binas et al. [47] evaluated the impact of miR-221/222 on cardiac electrical remodeling. Cardiac miRNA expression was analyzed in a mouse model with altered electrocardiography parameters and severe heart hypertrophy. They found out that increased miR-221/222 levels are associated with certain forms of hypertrophy mediated by upregulation of renin–angiotensin–aldosterone system. Another important finding is that alteration of miR-221/222 expression is associated with changed Cav1.2 and GIRK1/4 channel density resulting in slower propagation of action potential and possibly disturbed electromechanical coupling, thereby making the heart more vulnerable to arrhythmia, what is manifested on electrocardiogram (ECG) as a prolonged QT interval [47].

Further evidence indicates that miRNAs are involved in L-type Ca^2+^ current downregulation. There is the negative regulatory relationship between miRNA-29a-3p and the CACNA1Cgene that encodes alfa1-subunits of the cardiac L-type calcium channel. The expression of CACNA1C is controlled by miR-29a-3p. The miR-29a-3p levels are dramatically increased and CACNA1C mRNA levels decreased in atrial tissues with AF compared to those without AF. The atrial myocytes exposed to 30 nM or 60nM of miR-29a-3p mimics having significantly lower levels of the CACNA1C protein [50]. Different mechanisms of AF influenced by miRNAs are summarized in Table 1.

## 4. Potential of Circulating microRNAs as Biomarkers of AF

Although AF is the most common sustained arrhythmia, its diagnosis is often problematic due to possible asymptomatic and consequently subclinical form [51,52]. Detection of subclinical AF is highly challenging as only a minority of the patients are diagnosed during standard examinations with a 12-lead ECG or a 24 h ECG Holter monitor [53]. Documented AF causes 15% of ischemic strokes; however approximately 25% of ischemic strokes is of an unknown etiology [54]. It is assumed that undetected subclinical AF is responsible for these strokes [55]. There is also evidence that asymptomatic AF is associated with a higher incidence of strokes in comparison to symptomatic AF [56]. Since the standard ECG monitoring is not sufficient for AF detection, circulating biomarkers might be of a paramount importance in the diagnostic management.

MiRNA are highly stable in biofluids and, therefore, can serve as potential circulating biomarkers. In recent years, numerous miRNAs have been proposed as biomarkers for the diagnosis and prognosis of AF [57]. However, findings of different studies are inconsistent and not all miRNAs reported are actually important in the pathogenesis of AF [58].

Compared to the traditional biomarkers, such as NT-proBNP or cTn-hs, miRNAs do not have such robust evidence yet. The independent association of increased cTn-hs with mortality and the thromboembolic risk in patients with AF is well established [59,60]. Furthermore, evidence from meta-analyses demonstrates a relation between NT-proBNP concentration and the success of electrical cardioversion or the catheter ablation in establishing of sinus rhythm [45,61,62]. However, all these traditional biomarkers are lacking specificity for AF. Whereas they can be used as good prognostic markers of this arrhythmia, their screening capacity for AF is low [63].

One of the most studied miRNAs in AF is miR-150 [64]. Liu et al. described that plasma miR-150 levels from AF patients are substantially lower than in healthy people and a low level of miR-150 is significantly associated with AF. Moreover, miR-150 levels in platelets of patients with AF are significantly reduced and correlate positively to circulating levels of miR-150, indicating that plasma miR-150 is presumably from platelet secretion [65]. In line with these findings, the miRhythm study found correlation between circulating and heart tissue levels of miR-150 in patients with AF. Established chronic AF also shows decreased levels of circulating miR-150 [66]. Another interesting miRNA is miR-328. The support for a role of miR-328 in AF comes from the Framingham Heart Study; miRNA profiling was performed on the whole blood of 2467 participants. Four miRNAs—miR-328, miR-150, miR-331 and miR-28—were associated with prevalent AF. However, only association of low levels of miR-328 was significant after adjustments for age, sex and technical factors [58].

According to a more recent study, five important miRNAs, miR-29b, miR-328, miR-1-5p, miR-21 and miR-223-3p, could act as potential biomarkers for AF [67]. The most upregulated was miR-223-3p, which induces apoptosis via inhibition of the Activity Regulated Cytoskeleton Associated Protein (ARC) gene [68].

Postoperative AF is a common complication after cardiac surgery. A preexisting atrial substrate appears to be important in postoperative development of dysrhythmia, but its preoperative estimation is challenging. Rizvi et al. [69] performed a study to find such markers. Based on the results of their study, MiR-29a, -b and -c were significantly reduced in patients who subsequently developed postoperative AF compared with those who remained in SR.

Coagulation is closely related with AF and miRNAs were found to play an important role in regulating several hemostatic processes [70]. In the clinical practice, it is very important to distinguish a cardio-embolic stroke from an atherothrombotic stroke because both types of strokes require different treatment strategy. The antiplatelet therapy is used for atherothrombotic stroke while the anticoagulation is indicated for cardio-embolic stroke caused by AF [71]. In order to identify potential biomarkers that distinguish the two strokes, a profile of microRNAs expression was analyzed by Lai-Te Chen et al. [72]. In their study, they found 30 miRNAs with different expression, including 8 upregulated and 22 downregulated. From these differently expressed miRNAs, miR-15a-5p, miR-17-5p, miR-19b-3p and miR-20a-5p were significantly associated with cardio-embolic stroke. Furthermore, a relation is identified between 166 genes regulated by the mentioned miRNAs and diseases that contribute to cardio-embolic stroke, mainly atrial fibrillation, mitral valve stenosis and aortic dissection. Therefore, miR-15a5p, miR-17-5p, miR-19b-3p and miR-20a-5p are promising biomarkers for differentiating cardio-embolic stroke from atherothrombotic stroke. Table 2 summarizes miRNAs as potential diagnostic biomarkers.

## 5. Potential of miRNAs as Treatment of AF

Although there are several therapeutic options available, treatment of AF remains challenging, especially due to adverse effects of the drugs. Currently, there is no specific treatment of AF (besides anticoagulation) that decreases mortality [76].

Moreover, it was demonstrated that miRNAs have been involved in electrical and structural remodeling, which is responsible for AF etiology. Therefore, miRNAs can be used as new therapeutics of AF [77]. If a downregulation of miRNA causes a disease, there is a possible solution to use miRNA mimics to compensate this miRNA downregulation. Mimics are synthetic double-stranded oligonucleotides resembling the miR-duplex and, therefore, substituting (“mimicking”) their effects [78,79]. Upregulation of miRNA can be antagonized by anti-miRNA oligonucleotides (antimiRs). The antimiRs are complementary to the miRNA and enable its inhibition [80]. Locked nucleic acid, miRNA sponges, erasers or masks represent other possibilities of knockdown regulation [81,82].

Technologies used in miRNA therapies utilize lipid-based vehicles, various viral systems, biodegradable scaffolds, so-called “exosome-encapsulated” miRNAs, or light-induced antagomir activation [83].

The challenge consists in potential off-target effects, delivery system issues, and safety [83]. The most important concern about miRNA therapy arises from its potential of targeting multiple pathways. High load of miRNA mimics may interfere with physiological processes in non-targeted organs, or non-targeted pathways in the targeted tissue. Moreover, miRNA mimics could also interfere with normal gene regulation through competition with endogenous uptake of double-stranded RNA [84].

Several studies in animal models have been performed to demonstrate the use of miRNAs. Shan et al. [37] utilized a canine model of AF by nicotine administration. Administration of nicotine increased AF vulnerability via atrial fibrosis caused by significant upregulation of transforming growth factor-beta1 (TGF-β1), TGF beta receptor type II (TGF- β RII) accompanied by a significant decrease in the levels of miRNAs miR-133 and miR-590. Authors further found that transfection of miR-133 or miR-590 into atrial fibroblasts decreases TGF-β1 and TGF-β RII levels and collagen content. These effects were abolished by the antisense oligonucleotides against miR-133 or miR590 [85].

Loss-of-function mutation in the *zinc finger homeobox 3 gene* (ZFHX3) is associated with increased risk of AF. Cheng et al. [86] analyzed the differential miRNA expression profile of stable ZFHX3-KD, control HL-1 cells and explored the potential underlying signaling. MiR-133a-3p and miR-133b-3p were significantly downregulated and miR-184-3p, miR-195a-5p, miR-195a-3p and miR-574-3p were significantly upregulated in ZFHX3-KD cells compared with control cells. The deregulation of calcium homeostasis could contribute to cardiac arrhythmia pathogenesis. ECG recordings revealed that the miR-133a/b mimics reduced ZFHX3 KD-induced atrial arrhythmia in mice.

Li et al. [87] used AF rat models and rat cardiac fibroblasts with overexpressed or inhibited miR-10a to investigate the possible role of miR-10a mediated TGF-β1 in rats with AF. They found that downregulation of miR-10a inhibits collagen formation, reduces atrial structure remodeling and decreases proliferation of cardiac fibroblasts, which leads to the suppression of cardiac fibrosis in AF rats via inhibition of the TGF-β1 signaling pathway.

In the AF process, the effect of up-regulated MiR-328 has been demonstrated by reducing the density of the Ca^2+^ channel type L. This contributed to atrial electrical remodeling. Overexpression of miR-328 has a pro-arrhythmogenic effect, confirmed in animal models and the delivery of the antagomiR was able to reverse the arrhythmogenic phenotype [83].

The role of miRNAs potential in the treatment of AF was demonstrated in animal models. Before being introduced into clinical practice, safety concerns in humans need to be overcome [88,89]. Examples of miRNAs used for therapeutic purposes is summarized in Table 3.

## 6. Conclusions

miRNAs provide a new perspective in the pathophysiology of AF. Their presence in the circulation and the variation of levels reflecting pathophysiological processes make them potentially interesting diagnostic biomarkers and treatment targets for AF. Nevertheless, in order to become useful clinical tools, several limitations have to be solved. However, if the promising results of the pilot studies will be confirmed on a larger trial-based scale, miRNAs based diagnosis and therapy might improve current management of patients with AF.

## Figures and Tables

**Figure 1 ijms-21-02742-f001:**
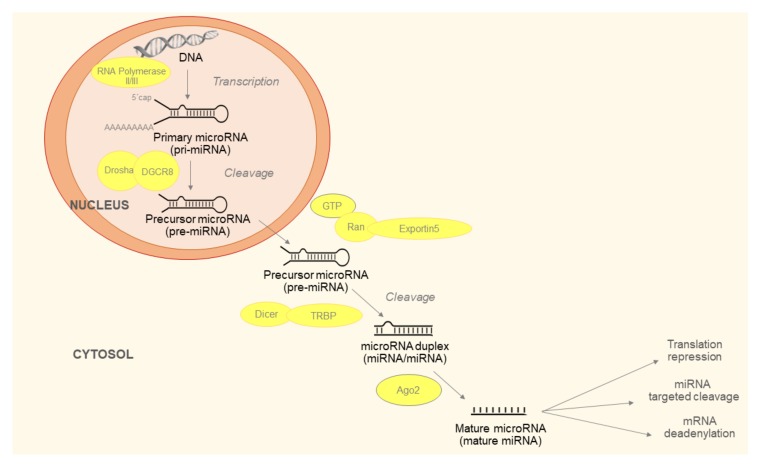
Micro ribonucleic acids (miRNAs) biogenesis begins with the production of the primary miRNA (pri-miRNA). It is transcribed by RNA polymerase II or III and cleavage of pri-miRNA by the microprocessor complex Drosha-DGCR8 (Pasha). The initial processing step occurs in the nucleus. In the next step, the pre-miRNA hairpin is exported to the cytoplasm by Exportin5/Ran-GTP complex. In the cytosol, the RNase III Dicer catalyzes the second processing step in complex with the double-stranded RNA-binding protein (TRBP), which cleaves the pre-miRNA hairpin to its mature length. Subsequently, the functional strand of the mature miRNA is incorporated together with Argonaute (Ago2) proteins to form the RNA-induced silencing complex (RISC). It inhibits target mRNA via cleavage, translational repression or deadenylation/degradation.

**Table 1 ijms-21-02742-t001:** Summary of microRNA (miRNA) mechanisms in atrial fibrillation.

Downregulated		Upregulated	
miRNAs	Mechanism	miRNAs	Mechanism
miR-132	Fibrosis via CTGF	miR-184-3p	Electrical remodeling via Cav1.2
miR-150	Fibrosis via c-myb	miR-195a-5p	Fibrosis via aniotensin II pathway
miR-1-5p	Fibrosis via TGF-β1/TGF-βRII	miR-195a-3p	Electrical remodeling via Cav1.2
miR-29a,b,c	Electrical remodeling via Cav1.2	miR-574-3p	Electrical remodeling via Cav1.2
miR-133	Fibrosis via TGF-β1 /TGF-βRII	miR-10a	Fibrosis via TGF-β1
miR-590	Fibrosis via TGF-β1 /TGF-βRII	miR-328	Electrical remodeling via Cav1.2
miR-21	Fibrosis via TGF-β1	miR-29a-3p	Electrical remodeling via Cav1.2
miR-132	Fibrosis via CTGF	miR-223p	Apoptosis via ARC
miR-221	Electrical remodeling via Cav1.2 and GIRK1/4	-	-
miR-222	Electrical remodeling via Cav1.2 and GIRK1/4	-	-

CTGF—the connective tissue growth factor, TGF-β1—transforming growth factor, β TGF-βRII—transforming growth factor beta receptor 2, CTGF—the connective tissue growth factor, c-myb—*C*-terminal domain of myeloblastosis family protein, eNOS—endothelial nitric oxide synthase, Cav1.2—subunit of L-type voltage-dependent calcium channel, GIRK—G protein-coupled inwardly-rectifying potassium channel, ARC—Activity Regulated Cytoskeleton Associated Protein.

**Table 2 ijms-21-02742-t002:** Overview of miRNAs as potential diagnostic biomarkers.

miRNA	Model	Reporting Studies	Results
hsa-miR-150	human	Goren et al. 2013 [65]	Significantly lower levels of circulating and platelet miR-150 in patients with atrial fibrillation (AF)
		miRythm Study 2015 [66]	2-fold lower miR-150 plasma in participants with AF than in those without AF 3-fold increase in plasma levels of miR-150 after AF ablation
cnf-miR-29b	canine	Dawson et al. 2013 [73]	Rapid decrease of miR-29b atrial expression in a canine congestive heart failure model of atrial fibrosis
hsa-miR-21	human	miRythm Study 2015 [66]	2-fold lower miRs-21 in plasma in participants with AF than in those without AF
		Da Silva et al. 2018 [16]	Lower expression of miR-21 in atrial tissue from patients with AF than in those without AF 3-fold increase in plasma levels of miRs-21 after AF ablation A significant increase in miR-21 plasma of patients with acute new onset of AF
hsa-miR-133b	human	Da Silva et al. 2018 [16]	A notable increase in miR-133b plasma of patients with acute new onset of AF
hsa-miR-328	human	Da Silva et al. 2018 [16]	An increase in miR-328 in plasma of patients with acute new onset of AF compared with patients with well controlled AF
		Lu 2015, [46]	Significant up-regulation of miRNA-328 in the atrial tissue of AF patients
hsa-miR-208a	human	Slagsvold et al. 2014 [74]	Increased expression of miR -208a in left vs. right atrium in tissue of patients with AF
hsa-miR-499	human	Da Silva et al. 2018 [16]	Notable increase in the expression of miR-499 in plasma of patients with a new onset of AF compared to patients in sinus rhythm
hsa-miR-328	canine	Lu et al. 2015 [46]	Significant up-regulation of miRNA-328 in the atrial tissue of experimental AF dogs
hsa-miR-1	human	Slagsvold et al. 2014 [74]	Increased expression of miR-1 in left atrium vs. right atrium in tissue of patients with AF
hsa-miR-142-5p	human	Wang et al. 2019 [75]	Lower expression of exosomal miR-142-5p in patients with AF compared to patients with sinus rhythm
hsa-miR-223-3p	human	Wang et al. 2019 [75]	Lower expression of exosomal miR-223-3p in patients with AF than in patients with sinus rhythm

hsa-miR—human micro ribonucleic acid, cnf-miR—canine micro ribonucleic acid, AF—atrial fibrillation, miR—micro ribonucleic acid.

**Table 3 ijms-21-02742-t003:** Overview of miRNAs as potential therapeutic targets.

miRNA	Model	Reporting Studies	Results
cnf-miR-133	canine	Shan et al. 2009 [37]	Downregulation of miR-133 and miR-590 associated with profibrotic effect of nicotine.
cnf-miR-590			
mmu-miR-133-a	rodent	Cheng et al. 2019 [86]	Cardiac remodeling and AF potentially reversed by MiR-133a/b mimics.
mmu-miR-133-b			
cnf-miR-206	canine	Zhang et al. 2015 [90]	Prolongation of atrial effective refractory period and reduction of AF inducibility by Anti-miR-206.
rno-miR-10	rodent	Li et al. 2019 [87]	Suppression of cardiac fibrosis in AF rats by MiR-10a down-regulation.
rno-miR-1	rodent	Yang et al. 2007 [91]	Anti-miR-1 reduced arrhythmogenesis in rat hearts after myocardial infarction.

cnf-miR—canine micro ribonucleic acid, mmu-miR—mouse micro ribonucleic acid, rno-miR—rat micro ribonucleic acid, AF—atrial fibrillation, miR—micro ribonucleic acid

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
