# Peer review of "Molecular Mechanisms, Diagnostic Aspects and Therapeutic Opportunities of Micro Ribonucleic Acids in Atrial Fibrillation"

_ijms, 2020, doi:10.3390/ijms21082742_

Round 1

Reviewer 1 Report

I read article entitled 'Molecular mechanisms, diagnostic aspects and therapeutic opportunities of micro ribonucleic acids in atrial fibrillation' with great interest.

This review paper concerns interesting topic: potential utility of micro ribonucleic acids (miRNAs) in the pathophysiology, diagnostics and treatment of atrial fibrillation (AF).

The title is brief and truly describes the core message of the paper. The abstract incorporates some key messages, but in a concise manner. The paper has sections on associations of AF with miRNAs well as potential underlying mechanisms. The general structure of the paper is accurate.
The paper contains details on methodology of discussed studies. On the other hand some relevant aspects of the scope of the paper seem omitted. The paper emphasizes future directions of research in the field of AF and miRNAs. I think this paper will be a valuable source of knowledge for researchers who are performing research in the field of AF.

However, I have some suggestions to further improve the paper with regards to presentation of scientific work and the scope of the paper:

  1. Abstract: miRNA will not be a diagnostic biomarker of AF – please rewrite the sentence – it could be rather a potential biomarker to provide the screening for AF or it may highlight a need to introduce more diagnostics directed towards AF – e.g. Holter ECG monitoring etc.
  2. Please include into table 1 data on miRNAs in AF obtained from miRBase (http://www.mirbase.org/), an online repository. Please make sure that the names of miRNAs are properly mentioned.
  3. The authors state “QPCR is relatively inexpensive and flexible (…)”. Please compare the cost of qPCR, NT-proBNP, troponin and GDF-15 for one patient in your laboratory. What is the percentage of hospitals in your country in which you may assess miRNA in clinical practice? This should be discussed.
  4. Please provide Table 2 (or provide more Tables if needed) with sections on miRNAs involved/potentially involved in the screening/diagnostics, risk stratification and therapy of AF – including potential studies on animal models (indicate this fact as a separate column or provide separate table[s] for animal studies).
  5. The authors state “One of the most studied biomarkers of AF is miR-150 [57].” Please compare this biomarker with other biomarkers studied in patients with AF, e.g. NT-proBNP, troponin and GDF-15 etc. 
  6. Were miRNAs studied in the context of the assessment of thromboembolic risk or prothrombotic state, e.g. fibrin clot properties?
  7. Please rewrite the sentence “By now there is no specific treatment of AF that decreases mortality [63]” highlighting the role of anticoagulation in patients with AF.

Author Response

Dear Reviewer,

thank you very much for the questions.

We have incorporated suggestions into Manuscript. Please, see answers to your suggestions.

  • Abstract: miRNA will not be a diagnostic biomarker of AF – please rewrite the sentence – it could be rather a potential biomarker to provide the screening for AF or it may highlight a need to introduce more diagnostics directed towards AF – e.g. Holter ECG monitoring etc.

Thank you for your comment and clarification of the term, the original text has been redesigned in the abstract. (Line 24).

  • Please include into table 1 data on miRNAs in AF obtained from miRBase (http://www.mirbase.org/), an online repository. Please make sure that the names of miRNAs are properly mentioned.

Thank you for bringing to the correct source of information at mirbase.org The additional information on the miRNA nomenclature has been added and can be found in line 46 (highlighted in red). and reported a correction in the names of particular miRNAs. Please, see Table 1 for the changes.

  • The authors state “QPCR is relatively inexpensive and flexible (…)”. Please compare the cost of qPCR, NT-proBNP, troponin and GDF-15 for one patient in your laboratory. What is the percentage of hospitals in your country in which you may assess miRNA in clinical practice? This should be discussed.

The reviewer asked us a very useful question. In line 124, we present a brief review of the cost comparisons of qPCR analysis for biomarkers: proBNP, troponin and GDF-15 in Slovakia. The challenge in diagnosing these biomarkers seems to be the availability of new diagnostic kits that could reduce the overall cost of performing effective diagnostics.

  • Please provide Table 2 (or provide more Tables if needed) with sections on miRNAs involved/potentially involved in the screening/diagnostics, risk stratification and therapy of AF – including potential studies on animal models (indicate this fact as a separate column or provide separate table[s] for animal studies).

Thank you very much for this comment, supplementing the above information on the diagnosis and therapy of individual miRNAs in relation to AF will be useful information that will complement the complexity of the diagnostic and therapeutic section. We have created Table 2 (Overview of miRNAs as potential diagnostic biomarkers) and Table 3 (Overview of miRNAs as potential therapeutic targets), which provide detailed information for each miRNA, including the sources we used.

  • The authors state “One of the most studied biomarkers of AF is miR-150 [57].” Please compare this biomarker with other biomarkers studied in patients with AF, e.g. NT-proBNP, troponin and GDF-15 etc. 

The reviewer asked us to compare biomarkers studied in AF patients (such as NT-proBNP, troponin and GDF-15). The answer is in line 246.

  • Were miRNAs studied in the context of the assessment of thromboembolic risk or prothrombotic state, e.g. fibrin clot properties?

This is a very interesting topic that was studied quite well. However, in order to fully address this topic, we would need to add several chapters that wouldn’t fit into the title of the manuscript. Nevertheless, we added a paragraph regarding the differentiation of atherothrombotic and thromboembolic stroke related to AF (273-284).

  • Please rewrite the sentence “By now there is no specific treatment of AF that decreases mortality [63]” highlighting the role of anticoagulation in patients with AF.

By the specific treatment of AF we meant pharmacological and interventional antiarrhythmic therapy however, we admit that it is confusing and we highlighted the role of anticoagulation (295-297).

Reviewer 2 Report

The review article by Allan Böhm and colleagues summarizes the potential clinical relevance and roles of microRNAs in atrial fibrillation well but is not comprehensive enough like few other previous reviews on the same topic (e.g., Cardiovasc Drugs Ther. 2017 Jun;31(3):345-365). At few places it is difficult to understand what authors wanted to convey and overall the review article requires comprehensive editing for language.

Few instances where editing and more information will improve the clarity of the message

  • There are currently discovered more than 2000 human miRNAs. Data on their structure and functions are collected in various online databases. The primary online repository represents miRbase [7]. Lines 46,47,48.
  • Due to number of miRNAs, a unique naming system was created. Line 49
  • The numerical indication can be followed by a letter, eg miR-133a or miR-133b, which almost doesn´t differ in structure from each other, but may differ in the targets, functions or in tissue expression. Lines 53,54,55. It is not clear what the authors mean by structure?
  • Compensate miRNA downregulation miRNA mimics are intensively investigating Mimics are synthetic double-stranded oligonucleotides resembling the miR-duplex and therefore ‘mimicking’ their effects [65,66]. Lines 264,265.
  • The table needs more examples and requires corresponding references within for miRNA examples mentioned.

Author Response

Dear Reviewer,

Thank you very much for your questions. We appreciate it.

Please read the answers:

  • There are currently discovered more than 2000 human miRNAs. Data on their structure and functions are collected in various online databases. The primary online repository represents miRbase [7]. Lines 46,47,48.

Thank you for your feedback on completing the information to improve the clarity of the information provided. Please see the line 46-51.

  • Due to number of miRNAs, a unique naming system was created. Line 49

Thanks for the warning, the information may sound confusing. For adjustment, see line 52-53.

  • The numerical indication can be followed by a letter, eg miR-133a or miR-133b, which almost doesn´t differ in structure from each other, but may differ in the targets, functions or in tissue expression. Lines 53,54,55. It is not clear what the authors mean by structure?

Thanks for your comment, we modified and justified the original claim (line 57-59).

  • Compensate miRNA downregulation miRNA mimics are intensively investigating Mimics are synthetic double-stranded oligonucleotides resembling the miR-duplex and therefore ‘mimicking’ their effects [65,66]. Lines 264,265.

We considered the reviewer's remark and adjusted the statement as follows – please see lines 294-297.

  • The table needs more examples and requires corresponding references within for miRNA examples mentioned.

The reviewer pointed out the need to supplement the information on the role of the miRNAs, which we also realized during the manuscript revision. We really appreciated this advice.  For this reason, we have included more detailed information on the diagnosis and therapy of miRNAs in Tables 2 and 3.

Round 2

Reviewer 1 Report

I read improved article entitled 'Molecular mechanisms, diagnostic aspects and therapeutic opportunities of micro ribonucleic acids in atrial fibrillation' with great interest.

The authors incorporated appropriate changes in response to my comments. However, I have some suggestions to further improve the paper with regards to presentation of the findings:

Abbreviations of atherothrombotic stroke (AS) or cardio-embolic stroke (CES) are not usual and seem to be used only in one paragraph. I think it is better to spell out.

The other introduced terms and abbreviations should be corrected as follows: growth differentiation factor-15 (GDF-15), N-terminal pro-B-type natriuretic peptide (NT-proBNP), high-sensitivity cardiac troponin (cTn-hs).

Author Response

Dear Reviewer,

thank you for your suggestions to further improve our manuscript.

Abbreviations of atherothrombotic stroke (AS) or cardio-embolic stroke (CES) are not usual and seem to be used only in one paragraph. I think it is better to spell out.

We have removed the abbreviations. Please see lines 275-285.

The other introduced terms and abbreviations should be corrected as follows: growth differentiation factor-15 (GDF-15), N-terminal pro-B-type natriuretic peptide (NT-proBNP), high-sensitivity cardiac troponin (cTn-hs).

We have corrected names and abbreviations of the biomarkers as you suggested. Please see lines 126-128 and 246-249.

Reviewer 2 Report

Authors have addressed the concerns in their revised version.

Author Response

Dear reviewer, thank you for your time and acceptance of the revisions.